# Genetic Alterations in Benign Adrenal Tumors

**DOI:** 10.3390/biomedicines10051041

**Published:** 2022-04-30

**Authors:** Georgia Pitsava, Constantine A. Stratakis

**Affiliations:** 1Division of Intramural Research, Division of Population Health Research, Eunice Kennedy Shriver National Institute of Child Health and Human Development, National Institutes of Health, Bethesda, MD 20892, USA; 2Section on Endocrinology and Genetics, Eunice Kennedy Shriver National Institute of Child Health and Human Development, National Institutes of Health, Bethesda, MD 20892, USA; stratakc@mail.nih.gov; 3Human Genetics & Precision Medicine, IMBB, FORTH, 70013 Heraklion, Greece; 4ELPEN Research Institute, ELPEN, 19009 Athens, Greece

**Keywords:** adrenal tumors, Cushing syndrome, PKA, *PRKAR1A*, genetics

## Abstract

The genetic basis of most types of adrenal adenomas has been elucidated over the past decade, leading to the association of adrenal gland pathologies with specific molecular defects. Various genetic studies have established links between variants affecting the protein kinase A (PKA) signaling pathway and benign cortisol-producing adrenal lesions. Specifically, genetic alterations in *GNAS*, *PRKAR1A*, *PRKACA*, *PRKACB*, *PDE11A*, and *PDE8B* have been identified. The PKA signaling pathway was initially implicated in the pathogenesis of Cushing syndrome in studies aiming to understand the underlying genetic defects of the rare tumor predisposition syndromes, Carney complex, and McCune-Albright syndrome, both affected by the same pathway. In addition, germline variants in *ARMC5* have been identified as a cause of primary bilateral macronodular adrenal hyperplasia. On the other hand, primary aldosteronism can be subclassified into aldosterone-producing adenomas and bilateral idiopathic hyperaldosteronism. Various genes have been reported as causative for benign aldosterone-producing adrenal lesions, including *KCNJ5*, *CACNA1D*, *CACNA1H*, *CLCN2*, *ATP1A1,* and *ATP2B3*. The majority of them encode ion channels or pumps, and genetic alterations lead to ion transport impairment and cell membrane depolarization which further increase aldosterone synthase transcription and aldosterone overproduction though activation of voltage-gated calcium channels and intracellular calcium signaling. In this work, we provide an overview of the genetic causes of benign adrenal tumors.

## 1. Introduction

Adrenocortical tumors (ATCs) originate within the adrenal cortex which makes up the outer portion of the adrenal gland. Histologically, the cortex has three distinct zones, zona glomerulosa (ZG), zona fasciculata (ZF), and zona reticularis (ZR). Each of the three zones has different functions depending on hormone production [1]. Due to the continuously increasing use of diagnostic imaging, adrenocortical lesions are being diagnosed more frequently than in the past. ATCs can be sporadic or familial, unilateral or bilateral, and secreting or non-secreting various adrenal steroids. Unilateral tumors are common in the general population, with prevalence 1–7% and often they are discovered by imaging studies intended to evaluate another disease; when discovered incidentally they are called incidentalomas [2]. The majority of them are benign adrenocortical adenomas (ACAs) and a small portion of them adrenocortical carcinomas (ACCs). ACAs are usually non-secreting; the long-term follow-up still remains controversial. According to the guidelines from the National Institutes of Health (NIH) [3] and the American Association of Clinical Endocrinologists (AACE/AASE) [4] dexamethasone suppression test should be repeated once a year for five years while imaging testing should be performed for at least one year if the tumor is <4 cm or for at least two years if the tumor is ≥4 cm. On the other hand, the European network for the study of adrenal tumors (ESE/ENSAT) and The European Society of Endocrinology suggest no follow-up studies if the initial presentation is typical of an adenoma [5]. 

However, 5–47% of ACAs can secrete cortisol leading to Cushing syndrome (CS) or in 1.6–3.3% they can secrete aldosterone and result in Conn adenomas [5,6]. 

On the other hand, even though ACCs are quite rare with estimated prevalence of 4–12 cases per million [7], they are responsible for steroid excess in 60–70% of cases [6,8,9]. Their prognosis varies depending on the tumor stage with 5-year survival rate ranging from 82% to 18% for tumors stage I and stage IV, respectively [5]. 

The genetic background of primary adrenal lesions has been unraveled through advances in the field of genomics over the past decade. The initial clues to our understanding came from a study of rare familial tumor syndromes and the identification of germline and somatic pathogenic variants in CS and primary aldosteronism (PA). These discoveries have facilitated the classification of adrenocortical lesions more accurately based on the causal gene while the genetic screening and counseling can be more individualized to each patient. 

In CS, aberrant cyclic adenosine monophosphate (cAMP)-protein kinase A (PKA) signaling has been found to be implicated in the majority of the benign cortisol-secreting ATCs [10,11]. The involvement of this pathway was first identified in CS in McCune-Albright syndrome (MAS), which is caused by somatic-activating variants in the gene that encodes the α-subunit of the stimulatory G protein (G_s_α), *GNAS* [11,12]. This was followed by identifying inactivating germline variants in the *PRKAR1A* in CS due to primary bilateral macronodular adrenal hyperplasia (PPNAD), which is part of Carney complex (CNC). Activating somatic *PRKACA* defects were discovered later as a major genetic defect of adrenal lesions producing cortisol. Furthermore, in the case of primary bilateral macronodular adrenal hyperplasia (PBMAH), germline defects in the tumor suppressor gene *ARMC5* have been discovered to be the primary cause [13]. In addition, in two other rare familial tumor syndromes associated with adrenocortical carcinomas, Li-Fraumeni and Beckwith–Wiedemann, defects in *TP53* and *IGF2* expression were identified respectively [14,15]. In the case of PA, its pathogenesis has been found to be linked to aberrant intracellular calcium signaling in aldosterone hypersecretion and defects in genes that encode ion channels such as *CACNA1H*, *CACNA1D*, *CLCN2*, *KCNJ5,* and ATPases including *ATP1A1* and *ATP2B3* have been implicated in adrenocortical tumorigenesis. This review aims to describe the causative molecular alternations in benign ATCs.

## 2. Benign Adrenocortical Tumors Producing Cortisol 

The incidence of CS is estimated to be 39–79 per million people per year in various populations with male to female ratio of 1:3 [16,17,18,19]. In 20–30% of cases of endogenous CS, it is caused by a primary adrenocortical process; 10–22% are caused by cortisol-producing adenomas (CPAs), adrenocortical hyperplasia which is mostly bilateral (BAH) in 1–2%, while ACCs are responsible for 5–7% of cases [16,20]. There are different forms of BAH, but the most common ones include ACTH-independent macronodular adrenal hyperplasia (AIMAH), PPNAD and isolated micronodular adrenocortical disease ((iMAD). In AIMAH, adrenocortical nodules have a diameter > 1 cm, whereas in the other two entities nodules are <1 cm. A genetic predisposition has been speculated because of their bilateral nature, which has been confirmed and a gene-based classification was proposed recently (Figure 1) [21,22].

The cAMP/protein kinase A (PKA) pathway plays a vital role in adrenocortical cell development, proliferation, and function. Normally, in adrenocortical cells, the pathway is activated by the adrenocorticotropic hormone (ACTH) that binds to its 7-transmembrane G protein-coupled receptor MC2R which activates Gs protein; that further increases cAMP levels and activated PKA (Figure 2). 

The PKA holoenzyme consists of four regulatory subunits RIα, RIβ, RIIα, RIIβ and three catalytic subunits Cα, Cβ, and Cγ and those form the type I and type II isoforms of PKA [24,25]. PKA is a hetero tetramer comprised of two regulatory subunits and two catalytic subunits. After binding of cAMP to PKA regulatory subunits the two catalytic subunits are activated and released and they phosphorylate various targets; that includes CREB (cAMP responsible element-binding protein), a transcription factor that is responsible for the stimulation of cAMP-dependent genes transcription. Since ACTH stimulates cell growth of the adrenal cortex and cortisol synthesis, it is reasonable to understand how irregular activation of the cAMP/PKA pathway is implicated in tumorigenesis of most benign cortisol-producing tumors and CS.

### 2.1. Micronodular Adrenocortical Hyperplasia

PPNAD is the most common form of micronodular adrenal hyperplasia and it is a rare cause of ACTH-independent hypercortisolism. It is most commonly diagnosed in children and young adults. PPNAD presents as part of CNC, a multiple neoplasia syndrome that consists of a complex of spotty skin pigmentation, myxomatous tissues of the heart, skin and other tissues, and endocrine tumors with or without overproduction of hormones. Endocrine tumors include PPNAD, pituitary adenomas, thyroid tumors, and others [26,27]. PPNAD is the most common endocrine neoplasm present in CNC patients as it occurs in approximately 25–60% of them [28,29]. 

CNC is inherited in an autosomal dominant manner. Germline inactivating variants in the *PRKAR1A* gene (17q22-24 locus) are the main cause of the disease and have been found in more than 70% of patients with familial CNC and PPNAD and more than 70% of patients with familial CNC with almost 100% penetrance [28,30,31]. *PRKAR1A* gene (17q24.2-24.3 locus- CNC1 locus) encodes the regulatory subunit type 1α (R1α) of PKA [28,30,31]. Inactivating defects lead to aberrant activation of the cAMP/PKA pathway. In the remaining cases, that do not harbor a germline defect in the *PRKAR1A* gene, genetic linkage analysis of tumors demonstrated another affected locus on chromosome 2p16 (CNC2 locus) [32,33]; however, the gene responsible has yet to be found. In a recent cohort of 353 patients with a germline *PRKAR1A* defects or a diagnosis of CNC and/or PPNAD, a genotype–phenotype correlation was performed; the study showed that the majority of patients with *PRKAR1A* defects and PPNAD harbored a germline c.709-7del6 variant while the remaining isolated PPNAD patients had the p.Met1Val alteration [28].

Rarely, somatic variants in *PRKAR1A* have been described in cortisol-producing adrenal tumors. In a cohort of patients with 44 sporadic adrenocortical tumors (29 adenomas and 15 cancers), losses in 17q22-24 were found in 23% of the adenomas and 53% of cancers, while inactivating variants in the *PRKAR1A* gene were identified in three tumors [34]. The *Prkar1a* knockout mice, in which the gene is specifically deleted in the adrenal cortex (AdKO), developed autonomous adrenal hyperactivity and bilateral hyperplasia resulting in BAH and CS [35].

More components of the cAMP/PKA pathway have been implicated over the years in the pathophysiology of PPNAD. Genetic alterations in the genes coding for the phosphodiesterases involved in cAMP degradation, *PDE11A,* which encodes phosphodiesterase type 11A and *PDE8B* which encodes phosphodiesterase type 8B, have been identified. A genome-wide SNP genotyping study in individuals with adrenocortical hyperplasia that was not due to known genetic defects (genetic alteration in *GNAS* or *PRKAR1A*) that included both leukocyte and tumor DNA was performed. The results of this study showed that variants in loci harboring PDE genes were most likely to be associated with the disease; inactivating variants in *PDE11A*, were found to be the most frequently linked, followed by the *PDE8B* gene [36]. In addition, in tumor specimens, the 2q31-2q35 (*PDE11A* was located there) locus was identified as the largest loss of heterozygosity region as well as increased cAMP-PKA signaling and CREB phosphorylation. In another study, *PDE11A* was sequenced in 150 patients with CNC that harbored germline *PRKAR1A* variants; interestingly, germline variants in *PDE11A* were significantly more frequent in CNC patients with PPNAD and/or testicular large-cell calcifying Sertoli cell tumors (LCCSCT) than in patients without PPNAD and/or LCCSCT. That could possibly suggest that *PDE11A* could act as a genetic modifying factor for the development of testicular and adrenal tumors in this population [37]. *PDE8B* locus was the second most likely region to be associated with a predisposition to PPNAD. A single base substitution c.914A>C/p.His305Pro was identified in a young girl with PPNAD that was diagnosed with CS at 2 years old, who inherited the pathogenic variant from her father. Consequently, in vitro studies in HEK293 cells demonstrated and confirmed the decreased activity of the mutant *PDE8B* [38]. 

However, genetic alterations in *PDE11A* and *PDE8B* have also been described in other kinds of adrenocortical tumors. A heterozygous-inactivating variant in *PDE11A* was identified in a non-secreting adrenocortical adenoma and heterozygous missense variants were more frequent in PBMAH (24%), ACA (19%), and ACC (16%) than in controls (5.7%) [39]. In an in vitro study by Vezzosi et al., it was confirmed that two *PDE11A* variants that were present in PBMAH and absent in controls, demonstrated decreased enzymatic activity compared to the wild-type [40]. In a case-control study, 216 adrenocortical tumors, negative for pathogenic variants in *PRKAR1A*, *GNAS,* and *PDE11A*, in unrelated patients and 192 controls were screened for genetic variations in *PDE8B*; six different variants in seven patients were identified (one PPNAD, one ACC, two PBMAH, two secreting-ACA, one non-secreting ACA) [41]. The deleterious effect to impair the protein function was confirmed for at least two of them [41].

In the recent years, genetic alterations in the catalytic subunits of the PKA enzyme have been found to play a role in micronodular BAH. A patient with CNC was found to have copy number gains on chromosome 1 of the *PRKACB* gene locus that encodes the catalytic subunit β (Cβ) of PKA. The patient presented with myxomas, acromegaly, and abnormal skin pigmentation; however, defects in *PRKACB* have not been linked to PPNAD. Interestingly though, the patient exhibited increased levels of Cβ in fibroblasts, lymphocytes, and breast myxoma and the increased lymphocytic cAMP-induced kinase activity was similar as in CNC patients with *PRKAR1A* defects [42]. *PRKACA*, that encodes the catalytic subunit α (Cα) of PKA has also been implicated in the development of IMAD. Germline copy number gains of the genomic region on chromosome 19p that includes the entire *PRKACA* gene were first described in two patients with familial PBMAH and in three patients with sporadic i-MAD. Tumor tissues from those patients had higher PKA Cα mRNA and protein levels with associated higher basal and cAMP stimulated PKA activity [10,43].

Another pathway speculated to be associated with the development of micronodular BAH is the wingless-type (*Wnt*)-β-catenin pathway. In this pathway, the Axin complex, which is comprised of Axin (a scaffolding protein), the *adenomatous polyposis coli* (APC), a tumor suppressor gene-, casein kinase 1 (CK1) and the glycogen synthase kinase 3 (GSK3), regulates the stability of β-catenin [44]. One study found somatic defects in the beta-catenin gene (*CTNNB1)* in two patients (11%) with PPNAD; one of the two patients harbored a germline *PRKAR1A* variant as well. These defects occurred in larger adrenocortical adenomas that developed in the background of PPNAD and were not present in the surrounding hyperplastic adrenocortical tissue [45]. In another study, immunohistochemistry and DNA sequencing were performed in PPNAD tissue (five with micronodules, three ACAs and one ACC the developed within an ACA) from nine patients (eight of them harbored *PRKAR1A* defects); accumulation of β-catenin was found in all PPNAD tissues while activating somatic *CTNNB1* variants were found in two of the five macronodules but were absent from the micronodules and the contralateral adrenal gland [46]. 

### 2.2. Macronodular Adrenal Hyperplasia (Pbmah)

PBMAH is characterized by bilateral adrenal macro-nodules with diameter > 1 cm. Multiple terms have been used to describe it: bilateral macronodular adrenal hyperplasia (BMAH), primary macronodular adrenal hyperplasia (PMAH), massive macronodular adrenocortical disease (MMAD), autonomous macronodular adrenal hyperplasia (AMAH), and “giant” or “huge” macronodular adrenal disease. The term ACTH-independent massive bilateral adrenal disease (AIMBAD) was also used in the past, but in studies in patients with PBMAH, cortisol secretion by the adrenals appeared to be regulated by corticotropin [47]. Rarely, it can cause adrenal CS (<2% of cases). Most frequently it appears to be sporadic or isolated; when hereditary it is transmitted in an autosomal dominant manner. In rare cases, it can also be unilateral and asymmetric. PBMAH is usually diagnosed in patients between the ages 40 and 65 that present with CS and low plasma ACTH levels or after an adrenal incidentaloma. 

Histologically, PBMAH is divided into two categories, type I PBMAH, which is characterized by internodular atrophic tissue, and the more common, type II PBMAH that exhibits diffuse hyperplasia and absence of normal or atrophic internodular tissue [48]. In the majority of PBMAH cases (ranging between 77 and 87% among studies), adrenocortical cells express aberrant hormone receptors, either excessive or ectopic [48,49,50]. These receptors are members of the GPCR family and are associated with steroidogenesis; thus, their stimulation significantly increases plasma cortisol. Such receptors include those for vasopressin, serotonin, angiotensin II, glucagon, glucose-dependent insulinotropic peptide (GIP), β-adrenergic agonists, and luteinizing hormone/choriogonadotropin (LH/hCG) [49,50,51,52,53,54,55,56,57,58]. Similar receptors have been found less frequently in adrenocortical adenomas and ACCs [48]. So far, despite whole-genome approaches, the genetic background causing the ectopic receptor expression has yet to be fully understood [59]. 

Increased signaling of cAMP-PKA pathway has been implicated in PBMAH, as it does in micronodular BAH. Genetic defects including inactivating germline variants in *PDE11A* (in 24–28% of cases) and *PDE8B*, *PRKACA* copy number gains as well as somatic *GNAS* defects without MAS have been described [39,40,41,48]. An isolated case of PBMAH has been reported, in which two variants (p.C21R and p.S247G) on the same allele of *MC2R*, encoding the melanocortin 2 receptor or ACTH, led to autonomous cortisol secretion through constitutive activation of the cAMP-PKA pathway [60]. It is interesting that either of the two defects alone, would result in inactivation of the receptor [60]. Defects in *PRKAR1A* have not been identified yet, but somatic losses of the 17q22-24 region in PBMAH cause the same alterations to PKA expression and activity as *PRKAR1A* defects or 17q losses [61]. The last component of cAMP-PKA pathway associated with PBMAH is the Gα subunit, which is encoded by *GNAS1* [62]. *GNAS1* activating variants cause MAS, a syndrome that manifests with “café au lait” spots, precocious puberty, polyostotic fibrous dysplasia, and hyperfunction of multiple endocrine glands [63,64]. The variants are somatic and cause constitutive activation of the cAMP-PKA which results in the formation of cortisol-producing adenomas [65]. 

Rare cases of PBMAH as part of other tumor predisposition syndromes have been described including familial adenomatous polyposis (*APC*), multiple endocrine neoplasia type 1 (*MEN1*), hereditary leiomyomatosis (fumarate hydrogenase, *FH*), and renal cell carcinoma (fumarate hydrogenase, *FH*) [66,67,68,69]. It is important to mention though that these comprise only a very small part of PBMAH cases and are associated with other tumors as well [67,69,70,71]. 

Later, inactivating variants in the *ARMC5* gene were discovered to be a genetic defect involved in the pathogenesis of PBMAH [13]. Genotyping of blood and tumor DNA from 33 patients with PBMAH, identified *ARMC5* (16p11.2 locus) variants in 18 of them (55%) [13]. All tumor samples of those 18 patients carried two genetic alterations in the *ARMC5* locus; however, their leukocyte DNA only carried one of the two suggesting that *ARMC5* is a tumor suppressor gene [13]. Subsequently, the prevalence of *ARMC5* germline variants in PBMAH was estimated to be 21–26% [72,73]. In one family carrying the p.A110fs*9 *ARMC5* defect, adrenal hyperplasia was associated with meningioma [74]. Additionally, an association between *ARMC5* and primary hyperaldosteronism was reported in 2015 [75]. Recently, a study in *armc5*-KO mice demonstrated a high rate of embryonic death and growth retardation; those who were older (>15 months) had CS and both the cAMP-PKA and the *Wnt*-β–catenin pathways were involved [76,77]. 

*ARMC5* is a tumor suppressor gene that encodes a cytosolic protein with no enzymatic activity that has an armadillo repeat domain, similar to the gene for β-catenin [76,78]. Proteins that contain armadillo domains are part of various functions including neural tube, T-cell and adrenal cortex development, and tumor suppression. Initially, functional investigations showed that inactivation of *ARMC5* was linked to decreased expression of steroidogenesis enzymes and in cortisol synthesis [13,73]. Thus, hypercortisolemia in those patients could be due to the increase in the number of adrenocortical cells [13]. Usually, patients carrying *ARMC5* deleterious variants tend to have overt CS, higher number of adrenal nodules, and larger adrenal glands [73]. 

A few other genes have been reported as possible culprits in a limited number of cases. Those include somatic variants in two genes involved in chromatin organization, histone modification, and thus regulation of gene transcription, *DOTIL* that encodes a histone H3 lysine methyl-transferase and *HDAC9* that encodes a histone deacetylase [79]. A single variant in *Endothelin Receptor type A EDNRA* gene which encodes a G-coupled protein, was reported in a study from two siblings from a family with familial PBMAH; however, it has not been confirmed in other studies yet [80].

### 2.3. Adrenocortical Adenomas Producing Cortisol

In CPAs, as in BAHs, the cAMP-PKA pathway predominates again. Yet, somatic genetic alterations are the most common defects leading to aberrant signaling of the pathway compared to BAH in which germline defects predominate [81]. The most prevalent defect involves somatic variants in *PRKACA* as identified in a WES in eight out of ten patients with CPAs; seven of them carried the same variant (c.617A>C/p.Lys206Arg) [10]. In addition, *PRKACA* variants were described in 22 of the 59 (37%) of unilateral adenomas. The variants were found only in patients with overt CS and the phenotype appeared to be more severe in those patients [10]. The c.617A>C/p.Lys206Arg variant was further described in four studies and the prevalence varied between 28 and 50% [79,82,83,84]. Activating variants of the PRKACA lead to constitutive activation of PKA by terminating the interaction between its catalytic and regulatory subunits as well as possibly by altering the specificity of the substrate by hyperphosphorylating certain substrates [85]. An activating somatic variant in *PRKACB* in a patient with CPA has also been reported recently; higher sensitivity to cAMP was demonstrated in in vitro studies [86].

Somatic inactivating defects in both *PRKAR1A* and *GNAS* have been identified in CPAs with prevalence to be estimated 5% and 4.5–11%, respectively [7,34,83,87,88,89]. Even though genetic alterations in *PRKAR1A* and *GNAS* increase signaling of the cAMP-PKA pathway, they seem to activate different downstream effectors. In a whole genome expression profile study, it was shown that adrenal lesions that harbored defects in *PRKAR1A* or *GNAS* resulted in overexpression of MAPK and p53 signaling pathways. In addition, in *GNAS*-mutant tumors, extracellular matrix receptor interaction and focal adhesion pathways (including *NFKB*, *NFKBIA,* and *TNFRSF1A*) were overexpressed, while in *PRKAR1A*-mutant tumors genes related to *Wnt*-signaling pathway, including *CCND1*, *CTNNB1*, *LEF1*, *LRP5*, *WISP1,* and *WNT3*, showed increased expression [90]. 

## 3. Benign Adrenocortical Tumors Producing Aldosterone (Adenomas and Hyperplasias)

Aldosterone is a steroid hormone that plays a vital role in regulating blood pressure by promoting reabsorption of sodium in the kidney. Primary aldosteronism (PA) is a heterogeneous group of disorders that is characterized by excess aldosterone and it is the most common form of secondary hypertension accounting for 5–10% of cases in primary care [91,92,93] and 20% of patients with resistant hypertension [94,95]. Aldosterone excess originates from adrenal glands, either from one or both and thus it can lead to unilateral or bilateral PA. PA is classified into two main subtypes, aldosterone-producing adenomas (APAs) and bilateral adrenocortical hyperplasia (BAHs), in 65% and 35% of cases, respectively [96]; the remaining cases include unilateral hyperplasia (2%), familial hyperaldosteronism (FH) (<1%), aldosterone-producing ACC (<1%), and ectopic aldosterone-producing adenoma or carcinoma (<0.1%) [97]. Under normal conditions, aldosterone synthesis is regulated by the renin-angiotensin-aldosterone system (RAAS) and potassium [98]. Moreover, it responds acutely to ACTH [99]. The two most important stimuli for the activation of RAAS include intravascular volume depletion and hyperkalemia. In the first case, angiotensin II is released and binds to a G-protein coupled receptor on adrenal glomerulosa cells while in the case of hyperkalemia, production of aldosterone from glomerulosa cells is direct [100]. Under those conditions, membrane depolarization and activation of voltage-gated calcium channels occur; intracellular calcium increases and leads to increase expression of aldosterone synthase (CYP11B2), aldosterone production and glomerulosa cell proliferation (Figure 3). Following these events, aldosterone acts on kidneys, on the mineralocorticoid receptor, specifically from the renal distal convoluted tubule to cortical collecting tubule where it increases potassium excretion and reabsorption of renal sodium.

### 3.1. Familial Hyperaldosteronism

FH is responsible for 1–5% of PA cases and it is inherited in an autosomal dominant manner [96]. It is subdivided into four forms, type I to type IV (Table 1) [96]. 

#### 3.1.1. Familial Hyperaldosteronism Type I

FH type I is also known as glucocorticoid-remediable aldosteronism (GRA); it was first described in 1966 in a case of a father and a son that presented with PA symptoms that were corrected with administration of glucocorticoids [101]. 26 years later, the molecular etiology of GRA was discovered [101,102]. GRA is a result of a chimeric gene that is formed by *CYP11B1*, which encodes 11β-hydroxylase that converts 11-deoxycortisol to cortisol, and *CYP11B2*, that encodes aldosterone synthase that catalyzes the conversion of deoxycorticosterone to corticosterone and 18-hydroxycorticosterone to aldosterone, both on chromosome 8. The chimeric gene leads in overproduction of aldosterone that is regulated by ACTH [102]. This hypersecretion can be reversed by intermediate-acting glucocorticoids [103]. 

#### 3.1.2. Familial Hyperaldosteronism Type II

Cases of FH type II were first reported in 1991 in a family that presented with APA and/or BHA unresponsive to glucocorticoids [104]. More familial cases were described after that [105]. A genetic cause was not known until recently that germline variants in *CLCN2* were identified in a study that included a family with FH II [106]. Additionally, 80 probands with early onset PA and without known variants were analyzed and several germline variants of the same gene were reported (9.9%) [106]. Around the same time, a study analyzing early onset PAs in 12 patients, found a *CLCN2* de novo germline variant [107]. *CLCN2* encodes the chloride channel CIC2, which is expressed among other tissues in the adrenal glands. Gain of function variants increase chloride efflux and depolarization of the plasma membrane resulting in influx of calcium [106,107]. Somatic variants in *CLCN2* have been reported in sporadic APA, however they are quite uncommon [108,109].

#### 3.1.3. Familial Hyperaldosteronism Type III

Whole exome sequencing was used to analyze 22 cases of APAs, in 2011, and two somatic variants in the *KCNJ5* gene (G151R and L168R) were identified in eight of them [110]. *KCNJ5* encodes a G-protein coupled inward rectifying potassium channel 4 (GIRK4). Defects in this gene alter the selectivity of the channel, lead to increased intracellular sodium influx and cell polarization, and thus increased intracellular calcium and calcium signaling [110]. Further studies that included ACC cell lines, demonstrated that after introduction of the *KCNJ5* variant, aldosterone synthesis was increased through the cell membrane depolarization and calcium and sodium influx [111,112,113,114]. Over the years, more *KCNJ5* variants have been reported [112,115,116,117,118,119,120,121,122,123,124,125,126,127]. A meta-analysis on somatic *KCNJ5* variants in patients with APA showed that they have more pronounced features of hyperaldosteronism, are more commonly young females, and their tumors are larger [128]. Regarding germline *KCNJ5* variants, they were first described in 2008 when a family presented with a new form of glucocorticoid- refractory PA [129]. This family was analyzed genetically a few years later and the *KCNJ5* variant was identified [110]. Following that, various phenotype-genotype correlations of FH III have been described ranging from more severe [110,130,131,132] to milder [130,133,134] cases. Recently, mosaicism for a *KCNJ5* defects in two cases of early onset PA was described [135,136].

#### 3.1.4. Familial Hyperaldosteronism Type IV

FH type IV is caused by germline defects in *CACNA1H* gene that encodes T-type calcium channels and it was initially described in 2015 [137]. Those defects caused early onset of PA in five out of 40 (12.5%) individuals; in two of them they were de novo events. Pathogenic variants in this gene impair channel inactivation and activation at more hyperpolarized potentials resulting in increased intracellular calcium levels [137]. Germline defects in *CACNA1D*, which encodes L-type calcium channels has also been found to cause PA; the difference is that those occur exclusively de novo, are not inherited from the parents and present with more severe phenotype, including seizures, neurological abnormalities (PASNA syndrome) [138]. Furthermore, germline pathogenic variants in *ARMC5* have been reported in patients with PA as well as germline variants in *PDE2A* and *PDE3B* [75,139]. The former two genes were associated with PA due to BAH but are not yet considered genetic causes of FH [139]. 

### 3.2. Aldosterone-Producing Adrenocortical Adenomas 

Almost all (90%) of APAs are due to somatic variants in genes encoding ion channels or transporters including *KCNJ5*, *CACNA1D, ATP1A1,* and *ATP2B3* [126,127,138,140]. 

The most frequent defects are in the *KCNJ5* gene and account for 40% of APAs; two particular variants (p.G151R and p.L168R) are responsible for the majority of those cases (36%). In addition, *KCNJ5* variants seem to be more common in females compared to males (53–63% vs. 22–31%) and more frequent in Asian cohorts (60–70% of APAs) than in European cohorts [120,127,128,141,142]. The next more common genetic defect includes somatic variants in the *CACNA1D* that accounts for up to 10% of APAs [126,127,138,143] and was found to be the most prevalent genetic defect in APAs among Blacks (42%) [123]. However, this is true only for Black males as among Black females KCNJ5 variants continue to have high prevalence [123]. 

A smaller percentage (3–17%) of APAs is caused by gain-of-function somatic variants in the ATPases *ATP1A1,* which encodes the α1-subunit of Na^+^/K^+^ ATPase and *ATP2B3*, which encodes the plasma membrane Ca^2+^ ATPase type 3 (PMCA3) [140,144]. The α1-subunit of Na^+^/K^+^ ATPase has ten transmembrane domains (M1–M10) and various variants (L104R, V332G, G99R, EETA963S) have been identified in the domains M1, M4, M9 [118,126,140]. Specifically, variants in the M1 and M4 domains, which compromise K^+^ binding, cause autonomous secretion of aldosterone driven by the depolarization of the cell membrane [140]. Variants in the M9 domain cause a loss of pump activity by damaging Na^+^-binding site [126]. The above-mentioned genetic alterations have been linked to abnormal H^+^ or Na^+^ leakage current which mechanistically resembles *KCNJ5* [126]. 

PMCA3, which transports Ca^2+^ out of the cell, also has ten transmembrane domains (M1–M10). Here, the majority of variants involved in APAs are deletions within a specific region of the M4 domain which is involved in binding of Ca^2+^ and ion gating [115,116,118,122,123,126,127,140,145,146]. In vitro studies have shown that *ATP2B3* variants increase production of aldosterone by decreasing export of Ca^2+^ (due to loss of the pump function) leading to increased intracellular concentration of Ca^2+^, membrane depolarization, and activation of calcium signaling [147,148]. 

Finally, activating somatic variants in *CTNNB1* have been reported in 2–5% of APAs with a high portion of them exhibiting constitutive activation of the Wnt-β-catenin pathway [115,138,149,150,151]. Those variants have also been described in two females with APAs that presented in pregnancy with increased adrenocortical expression of the LH/hCG receptor and gonadotropin releasing hormone (GnRH) receptor [152]. However, this association was not confirmed in a subsequent study [153].

## 4. Conclusions

Significant advances have been made in the recent years that led to better understanding of the molecular background of adrenocortical tumors. The dysregulation of the cAMP-PKA signaling pathway is vital in the development of those tumors. These advances are very important in transforming them to new diagnostic and therapeutic targets.

## Figures and Tables

**Figure 1 biomedicines-10-01041-f001:**
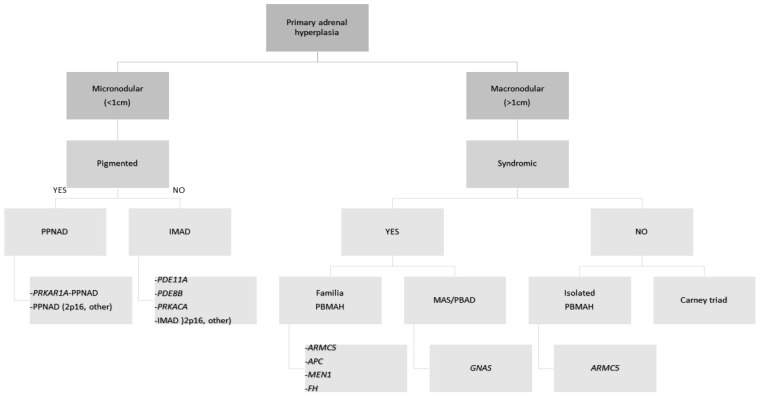
Algorithm for the diagnosis of primary cortisol-producing adrenocortical hyperplasias based on the underlying genetic etiology. *APC* adenomatous polyposis coli gene, *ARMC5* armadillo repeat-containing protein 5, *c-PPNAD* CNC-associated primary pigmented nodular adrenocortical disease, *CNC* Carney complex, *FH* fumarate hydratase, *GNAS* gene coding for the stimulatory subunit α of the G-protein (Gsα), *i-MAD* isolated micronodular adrenocortical disease, *i-PPNAD* isolated PPNAD, *MAS* McCune–Albright syndrome, *MEN1* multiple endocrine neoplasia type 1, *PBAD* primary bimorphic adrenocortical disease, *PBMAH* primary bilateral macronodular adrenocortical hyperplasia, *PDE8B* phosphodiesterase 8B gene, *PDE11A* phosphodiesterase 11A gene, *PPNAD* primary pigmented nodular adrenocortical disease, *PRKACA* protein kinase cAMP-dependent catalytic, alpha, *PRKAR1A* protein kinase cAMP-dependent regulatory type Iα gene. Adapted from Kamilaris CDC, Stratakis CA, Hannah-Shmouni F, 2020 [23].

**Figure 2 biomedicines-10-01041-f002:**
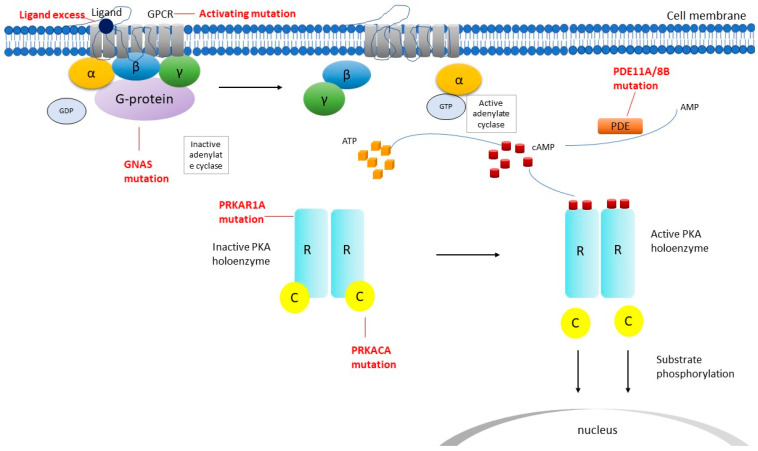
Schematic representation of cyclic adenosine monophosphate (cAMP) signaling pathway in primary adrenal neoplasms. *C* catalytic subunit of PKA, *GDP* guanosine diphosphate, *GNAS* gene coding for the stimulatory subunit α of the G-protein (Gsα), *GPCR* G-protein coupled receptor, *GTP* guanosine triphosphate, *PDE* phosphodiesterase, *PDE11A/8B* phosphodiesterase 11A and 8B respectively, *PKA* protein kinase, *R* regulatory subunit of PKA, *α*, *β*, *γ* subunits, *PRKACA* protein kinase cAMP-dependent catalytic, alpha *PRKAR1A* protein kinase cAMP-dependent regulatory type Iα gene.

**Figure 3 biomedicines-10-01041-f003:**
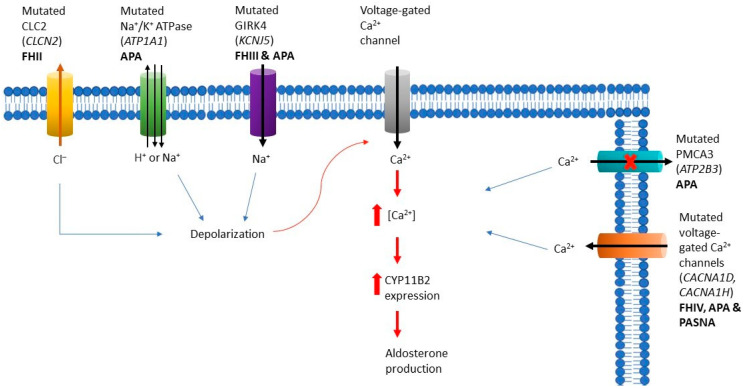
Cellular mechanisms leading to aldosterone production in aldosterone-producing adenoma and familial hyperaldosteronism. Genetic alterations in *CLCN2*, *ATP1A1,* and *KCNJ5* result in defective ion transport and thus membrane depolarization, which in turn activates voltage-gated Ca^2+^ channels and increases Ca^2+^ levels. Genetic alterations in *ATP2B3* decrease export of Ca^2+^, while in *CACNA1D* and *CACNA1H* directly increase Ca^2+^ levels. Increased intracellular Ca^2+^ enhances expression of aldosterone synthase (CYP11B2) and promotes aldosterone production. *APA* aldosterone-producing adenoma, *CLC2* chloride channel 2, *FH* familial hyperaldosteronism, *GIRK4* G-protein coupled inward rectifying potassium channel 4, *PMCA3* Ca^2+^ ATPase type 3.

**Table 1 biomedicines-10-01041-t001:** Molecular and clinical characteristics of familial hyperaldosteronism (FH) [96].

Familial Hyperaldosteronism	Gene	Clinical Characteristics
Type I	*CYP11B1/CYP11B2* chimeric gene	Glucocorticoid-suppressive hyperaldosteronism
Type II	*CLCN2*	Early onset PA
Type III	*KCNJ5*	Severe early-onset PA (T158A, I157S, E145Q, G151R) Mild PA (G151E, Y152C)
Type IV	*CACNA1H*	Early onset PA

*PA* primary aldosteronism.

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
