# Peer review of "Genetic Alterations in Benign Adrenal Tumors"

_biomedicines, 2022, doi:10.3390/biomedicines10051041_

Round 1
Reviewer 1 Report
The authors provide a very nice and comprehensive view on the genetics in benign adrenal tumors. Especially the section on cortisol producing tumors is very detailed. In comparison to that section of the manuscript, the second part on primary aldosteronism seems a little less comprehensive. Therefore, I would suggest to go a little bit more into detail especially regarding classical APAs and respective mutations, maybe add an additional figure on the altered channel/pump functions. Furthermore, I feel, that a section on non-functioning adenomas is lacking.
Reviewer 2 Report
This minireview describing the genetic aberrations in benign tumors is thoroughly referenced and makes a significant contribution to endocrinology.
The topic is original and encompasses the genetic causes of adrenal tumors seen in the various subtypes of primary aldosteronism and Cushing syndrome.
The genetic landscape of the various types of adrenal adenomas has been elucidated in the past decade. The current manuscript highlights the genetic basis of adrenal adenomas that are seen in primary aldosteronism and Cushing syndrome. The research is well-referenced and interesting.
Author Response
We sincerely thank Reviewer 2 for his/her nice comments.